# Factors Associated with Tooth Loss in Postmenopausal Women: A Community-Based Cross-Sectional Study

**DOI:** 10.3390/ijerph16203945

**Published:** 2019-10-16

**Authors:** Mei-Yu Pan, Tsung-Cheng Hsieh, Po-Han Chen, Mei-Yen Chen

**Affiliations:** 1Department of Nursing, Chang Gang University of Science and Technology, Chiayi 61363, Taiwan; pmeiyu@gw.cgust.edu.tw; 2Institute of Medical Sciences, Tzu Chi University, Hualien 97004, Taiwan; tchsieh@mail.tcu.edu.tw; 3Department of Orthopedic Surgery, Chang Gung Memorial Hospital, Yunlin 63862, Taiwan; chaos5408@gmail.com; 4College of Nursing, Chang Gung University of Science and Technology, Chiayi 61363, Taiwan; 5Department of Cardiology, Chang Gung Memorial Hospital, Chiayi 61363, Taiwan; 6School of Nursing, Chang Gung University, Taoyuan 33303, Taiwan

**Keywords:** bone mineral density, number of remaining teeth, oral hygiene, postmenopausal

## Abstract

Many studies have indicated that menopause affects periodontal health and tooth loss. The possible mechanism might due to several hormonal changes and low bone mineral density (BMD) during the transition period. However, few studies have explored the role of oral hygiene in the number of remaining teeth (NRT) in postmenopausal women (PMW). The aim of this study was to explore the prevalence of and factors associated with NRT less than 20 in PMW. A community-based, cross-sectional study was conducted in coastal Yunlin County, Taiwan. NRT was calculated based on natural and filled teeth. BMD was detected by dual-energy X-ray absorptiometry at the collaborating hospital. Logistic regression analyses were used to identify the factors associated with NRT in rural PMW. Six hundred and ten rural PMW with mean age 65.4 years enrolled in this study. The mean NRT was 17.6 (standard deviation [SD] = 10.4), with 43.9% having <20 and 13.9% edentulous. More than half (65.9%) reported that they seldom brushed their teeth after meals, 79.2% rarely used dental floss, and 80% did not regularly undergo tooth scaling by a dentist. The majority of women had low BMD, including 48.7% with osteopenia and 20.7% with osteoporosis. After adjusting for potentially confounding variables, NRT <20 was associated with infrequent tooth scaling (odds ratio [OR] = 2.78, 95% confidence interval [CI] = 1.70–4.56) and dental floss use (OR = 2.01, 95% CI = 1.24–3.26), but not BMD. A high prevalence of NRT <20 was found among rural PMW, but poor oral hygiene rather than low BMD was the major contributing factor. It is an emerging issue for primary healthcare providers and clinicians to initiate oral hygiene promotion programs for these disadvantaged women.

## 1. Introduction

Menopause is a period in which women’s health status is significantly fluctuating, with an average onset age of 50. The global average life expectancy for women is 74 years [1], so the time after menopause occupies nearly one-third of women’s lives. The secretion of estrogen rapidly declines post menopause, and leaves postmenopausal women (PMW) at higher risk of various physical and mental illnesses than men [2,3]. This imbalanced health status and postmenopausal reduction of estrogen has been blamed for numerous ailments, including tooth loss [4,5]. Retaining at least 20 natural teeth until 80 years of age has been defined as a key element of proper general health [6]. However, many studies on tooth loss have focused on low bone mineral density (BMD) and periodontitis in PMW but less on the oral hygiene behaviors owned by those with <20 remaining teeth [7,8].

In recent years, osteoporosis has received much attention on the issue of oral health in PMW [9]. Studies have indicated that osteoporosis, considered to be a problem of bone destruction and associated with rapid fluctuations in estrogen, exaggerates the loss of alveolar bone, which is involved in the retention of teeth [10,11,12]. In fact, there are many important factors associated with tooth loss and osteoporosis, such as advanced age, poor education [13,14,15], chronic diseases [15,16], tobacco and alcohol use [13,16,17], oral hygiene, and unhealthy diet [16,17,18]. However, there is still controversy in the literature on the relationship between BMD and tooth loss in PMW [9,16].

Oral hygiene is generally ignored in the relationship between osteoporosis and tooth loss in PMW [8,12,16]. Oral hygiene behaviors, such as brushing [19], flossing [17,20,21], regular tooth scaling and dental visits [22,23], have been highly recommended as the essential elements for retaining natural teeth and maintaining oral health. Hence, we doubt that hormone reduction or osteoporosis can be the only factors related to a number of remaining teeth (NRT) <20 in PMW. This study aimed to explore the prevalence and factors related to NRT <20 among rural PMW and to consider the role of BMD and oral health behaviors.

## 2. Materials and Methods 

### 2.1. Design, Sample, and Setting

This cross-sectional descriptive study was part of a community-based health promotion program for residents living in the rural coastal region of Taiwan. It was conducted by nursing faculties from October 2015 to July 2016 in collaboration with the local hospitals. This study used purposive sampling to select participants. The inclusion criteria were: (1) women with postmenopausal stage (identified by the physicians of the research team); (2) age ≥50 years and less than 80 years; (3) able to walk to the local hospital; (4) able to complete questionnaires independently or through interview; and (5) agree to participate and sign the informed consent before enrolling in the study. The exclusion criteria were: (1) unable to answer the questionnaires clearly; and (2) having serious disease(s), such as kidney failure that required regular dialysis, or cancer. 

### 2.2. Measurements

Demographic characteristics included age, years of education, and three self-reported chronic diseases (diabetes mellitus, hypertension, and hyperlipidemia).Bone mineral density (BMD) was measured with dual-energy X-ray absorptiometry at the collaborating hospitals. Participants with a T-score <−1.0 were classified as having osteopenia, and a T-score <−2.5 indicated osteoporosis, as recommended by WHO guidelines [24].Number of remaining teeth (NRT) was calculated by the trained research assistants based on the natural and filled teeth in the mouth. Removable dentures and residual roots without a crown were excluded. Participants were divided into two categories, namely “NRT <20” and “NRT ≥20”.Oral health related behaviors were measured by personal habits including oral hygiene, substance use, and eating behaviors. Firstly, oral hygiene was evaluated by three questions: “Do you use dental floss daily?”, “Do you undergo dental checkups or tooth scaling regularly (every 6–12 months can be covered by the National Health Insurance)?”, and “Do you brush your teeth after meals?” Answers were categorized as infrequent (“never/seldom”) versus frequent (“usually/always”). Secondly, tobacco and alcohol use were classified as “never used” versus “currently or formerly used”. Thirdly, eating behaviors were examined with two questions: “Does your diet include five food groups per day, namely grains, vegetables, milk, meat, and fruits?”, and “Do you drink at least 1500 mL of water every day? The answers “never or sometimes” were classified as inadequate, whilst “usually or always” were coded as adequate.

### 2.3. Procedure and Ethical Considerations

The study received ethical approval from the Institutional Review Board (No. 104-9925B). The research assistants were senior nursing students who had received 8 hours of training from the investigators for how to count the NRT. All of the instruments had good reliability (inter-rater reliability was 0.9) and validity (content validity was 0.88–0.89). Details of the interviews and measurement procedures were based on the published literature [13,15,17].

### 2.4. Statistical Analyses

The Chi-square and *t*-tests were used to examine the univariate factors associated with NRT. To identify the correctable factors related to NRT <20, a 3-level multivariate logistic regression with forward entry method was performed. Model 1 was used to detect the correlation between each single variable and NRT <20 without adjustment. Model 2 separately analyzed each modifiable factor by adjusting two irreversible confounding factors - age and education. Model 3 was a complete model considering all of the study variables to identify those associated with NRT <20. Collinearity diagnosis was performed by the correlation analysis between these category factors by using the Cramér’s V coefficient. The factor which was highly correlated with other factors was excluded from the model. The odds ratios (OR) with 95% confidence intervals (CI) were obtained. SPSS 22.0 software (IBM SPSS Statistics for Windows, Version 22.0. Armonk, NY, USA) was used and *p* < 0.05 was set as statistical significance.

## 3. Results

Among 651 postmenopausal women who were enrolled in this study, 41 failed to complete the NRT measurement, and 610 participants were evaluated, with an average age of 65.4 years (SD = 7.6), range 50–80. The mean years of education received was 3.1 (SD = 4.3). The prevalence of hypertension, diabetes, and hyperlipidemia was 46.6%, 47%, and 20.8%, respectively. Participants also had a high prevalence of low BMD, with mean T-scores of −1.6 ± 1.1, and 48.7% of participants having osteopenia, and 20.7% having osteoporosis. In addition, nearly 80% of participants reported seldom or never undergoing tooth scaling or using dental floss. Tooth brushing behavior after meals was also unsatisfactory (65.9% seldom or never). Regarding diet, 26.9% reported they did not eat five food groups daily, and 53.4% did not drink at least 1500 mL of water daily. The mean NRT was 17.6 (SD = 10.4) and 43.9% of the participants had NRT <20, including 13.9% who were edentulous (Table 1).

Univariate analysis showed that women who were older (>65 years old, *p* < 0.001) and had little education (*p* < 0.001), diabetes (*p* < 0.05), low BMD (*p* < 0.001), and osteoporosis (*p* < 0.01) tended to have NRT <20 (Table 2). Regarding the oral health related behaviors, infrequent tooth scaling (*p* < 0.001), infrequent dental floss use (*p* < 0.001) and cigarette smoking (*p* < 0.05) were significantly associated with NRT <20. Logistic regression analysis for Model 1 showed that diabetes, osteoporosis, and smoking were associated with NRT <20 (Table 3). However, the effect of these variables became insignificant after the adjustments in Models 2 and 3. In Model 3, a complete model adjusted for all variables, infrequent tooth scaling (OR = 2.78, 95% CI = 1.70–4.56) and infrequent dental floss use (OR = 2.01, 95% CI = 1.24–3.26) were significantly associated with the increased risk of NRT <20. The Cramér’s V coefficients between each pair of explanatory variables were less than 0.3 (not shown in the table) and showed low-degree correlations. Thus, all the explanatory variables were included in the logistic regression analyses.

## 4. Discussion

A nurse-led health promoting program was a feature of this study and the purpose was to promote oral health in rural PMW. This study had four main findings. First, PMW had a reduced number of teeth, with a high prevalence of NRT < 20 and edentulism. In previous studies, the average NRT for women aged 20 to 64 was 24, only 18.4% had NRT < 20 [17], and the average NRT for PMW was 22–23 [10,14]. The present study indicated that the oral health of these participants was significantly worse. In addition, the prevalence of NRT < 20 was not only higher than previously reported for women under the age of 65 [17], but also higher than that of PMW in Korea (25.9%) [5], and the United States (5.9%) [7].

Second, the oral hygiene behaviors of participants were generally poor, and far from the recommendations of the Fédération Dentaire Internationale (FDI) World Dental Federation [25]. In previous studies [17,19,23], the rare use of dental floss, never or seldom undergoing tooth scaling, and lack of tooth brushing after meals were significantly associated with NRT <20. The high prevalence of inadequate oral hygiene habits in PMW in this study was similar to a study in Korean PMW (67–81%) [16], but much higher than that found in the general population (58–70%) [17,21], or in PMW in other countries (28–52%) [7,12].

Third, the present study found that osteopenia or osteoporosis was not the major factor in increasing likelihood of NRT < 20. The study revealed that while the prevalence of osteopenia (48.7%) was similar to some other countries, the prevalence of osteoporosis (20.7%) was lower than other countries (24–38%) [10,12,14]. These phenomena might be due to sample variation, such as age, occupation, and geography, but the common feature is that PMW generally have a low BMD (prevalence >60%). 

Many reports indicate that systemic BMD might be associated with alveolar bone loss or tooth loss, due to rapid imbalance of estrogen levels affecting the activation of bone cells and immune cells [10,12]. However, the present study demonstrated that osteopenia or osteoporosis was not a significant factor in NRT < 20 in PMW. Dental health interventions in PMW should focus more on the impact of periodontal disease as a main cause of tooth loss in adults [18]. Periodontal disease is a product of cellular immune responses to oral inflammatory processes, which are caused by oral infections or estrogen deficiency, and is a common oral problem in PMW [10,11,26]. Further, the teeth and alveolar bone have a higher chance of infection and inflammation than other bones. Based on the principle that prevention is better than treatment, reducing the impact of inflammatory reactions on NRT <20 is worth more attention in PMW.

Good oral hygiene habits have been recognized as reducing oral inflammatory reactions [20,22]. Fortunately, the fourth finding indicated that regular tooth scaling and use of dental floss were two correctable factors related to NRT. This result is corroborated by studies in different populations [15,21,23] as well as a study in Korean PMW [5]. However, in some studies evaluating the relationship between osteoporosis and NRT in PMW, these two important oral hygiene behaviors were not considered [8,14]. A large study in South Korea [16] explored the loss of more than eight teeth in PMW and claimed that low bone density was associated with tooth loss in this population. Although the researchers considered many factors, such as BMD and oral care-related behaviors, they did not adjust for all variables in their analysis model. In summary, it is impossible to eliminate the contribution of oral health behaviors in retaining more natural teeth.

In fact, evidence indicates that dental flossing, regular dental checkups, and tooth scaling are the most effective ways to preserve NRT [18]. This is even more important in PMW, since the decline of estrogen during menopause reduces the ability to maintain the balance between beneficial and harmful bacteria in the oral environment, and the plaque biofilm accumulated by oral bacteria strengthens the inflammatory response in the body, increasing susceptibility to progressive periodontal disease. Consequently, the risk of tooth loss [4,11]. The best way to reduce this susceptibility is to remove the plaque under the gum-line by regular oral care [11,22,23]. Previous literature has also proved that dental plaque and gingivitis can be significantly improved by brushing and flossing compared with brushing alone [20], and even using interdental brush is superior to dental floss [27]. However, women were less concerned about oral health when compared with menopausal symptoms, osteoporosis, and other chronic diseases [3,28]. In this study, there were 65.9–80% of PMW who never or seldom used dental floss, brushed their teeth after meals, or underwent tooth scaling, even though this is covered by the national health insurance system every 6–12 months. In order to encourage regular use of dental health services, the insured is only required to pay minimal registration and co-payment fees (3–5 USD). Perhaps many women in the study population believe that osteoporosis and periodontal disease are inevitable in the postmenopausal stage [29], which might be misunderstood that the tooth loss in PMW is the fate of high susceptibility to osteoporosis, thus their reliance on excessive medical services to retain teeth and lack of awareness of the benefits of oral hygiene. Further studies are necessary to explore the prevailing level of knowledge and attitudes toward use of dental health services. 

Good oral health is an indicator of general health, and can reduce medical expenses and improve nutrition, communication, self-esteem, and quality of life [24,25]. Oral hygiene education could contribute significantly to improving oral health. It is urgent to strengthen the practice of dental flossing and regular tooth scaling among PMW to achieve the global goal of retaining at least 20 teeth through age 80 [25]. 

### Limitations

The advantages of this study were the discovery of the high prevalence of NRT < 20 and the importance of oral hygiene for maintaining NRT in PMW, but there were still some limitations. First, hormone therapy and osteoporosis medication data were not collected. Some studies have pointed out that estrogen use can prevent from tooth loss in PMW [5,14], while the widely used bisphosphonate drugs for the treatment of osteoporosis have been reported to have potential risks of osteonecrosis of the jaw [30]. According to other surveys, only about one-third of Taiwanese PMW have used or are currently treated with hormones or prescription drugs for osteoporosis [13,31]. Therefore, women’s oral health may be worse than the survey indicated in this study, if up to one-third of the participants are assumed to receive some benefit from estrogen therapy. Secondly, the trained research assistants only counted the number of teeth, and there was no assessment of other oral parameters, such as dental caries, periodontal disease, oral hygiene index, and plaque index. Therefore, the NRT data reported here do not address other potential oral problems. Finally, the recruitment of a single region and non-random sampling might limit the generalization of the findings.

## 5. Conclusions

For postmenopausal women in rural areas in Taiwan, the prevalence of NRT < 20 and edentulism is significantly higher than that reported in previous studies in other regions. Poor oral hygiene behavior was found to be common. The occurrence of NRT < 20 in postmenopausal women was most strongly related to infrequent flossing and tooth scaling, rather than age, education level, or osteoporosis. Low BMD was not a primary factor in NRT < 20, and tooth loss cannot be strictly attributed to hormonal changes in menopause. Awareness of oral hygiene is the most important issue for retaining dentition in postmenopausal women.

## Figures and Tables

**Table 1 ijerph-16-03945-t001:** Demographic characteristics of the participants (*N* = 610).

Variable	*n* (%)	Mean ± SD
Age, years		65.4 ± 7.6
≥65	321 (52.6)	
Education level, years		3.1 ± 4.3
≤9	564 (92.5)	
Chronic diseases		
Hypertension	284 (46.6)	
Diabetes mellitus	287 (47.0)	
Hyperlipidemia	127 (20.8)	
BMD T-score		−1.6 ± 1.1
BMD status		
Normal	187 (30.7)	
Osteopenia	297 (48.7)	
Osteoporosis	126 (20.7)	
Oral health related behaviors		
Tooth brushing after meal (infrequent)	402 (65.9)	
Dental flossing (infrequent)	483 (79.2)	
Tooth scaling (infrequent)	488 (80.0)	
Inadequate nutrition	164 (26.9)	
Inadequate water intake	326 (53.4)	
Cigarette smoking	12 (2.0)	
Alcohol drinking	36 (5.9)	
Number of remaining teeth (NRT)		17.6 ± 10.4
Group of NRT		
0	85 (13.9)	
1–19	183 (30.0)	
≥20	342 (56.1)	

SD, standard deviation; BMD, bone mineral density; NRT, number of remaining teeth.

**Table 2 ijerph-16-03945-t002:** Univariate analysis of factors associated with number of remaining teeth (N = 610).

Variable	NRT ≥20 (*n* = 342)	NRT <20 (*n* = 268)	*p*
Age, years (Mean ± *SD*)	63.3 ± 7.4	68.0 ± 6.9	<0.001
≥65	140 (40.9)	181 (67.5)	<0.001
Education years (Mean ± *SD*)	4.0 ± 0.2	2.0 ± 3.6	<0.001
≤9	306 (89.5)	258 (96.3)	0.002
Chronic diseases			
Hypertension	150 (43.9)	134 (50.0)	0.131
Diabetes mellitus	147 (40.3)	140 (52.2)	0.023
Hyperlipidemia	72 (21.1)	55 (20.5)	0.873
BMD T-score (Mean ± *SD*)	−1.4 ± 1.1	−1.7 ± 1.2	<0.001
BMD status			0.006
Normal	117 (34.2)	70 (26.1)	
Osteopenia	169 (49.4)	128 (47.8)	
Osteoporosis	56 (16.4)	70 (26.1)	
Oral health related behaviors			
Tooth scaling infrequent	246 (71.9)	242 (90.3)	<0.001
Using dental floss infrequent	244 (71.3)	239 (89.2)	<0.001
Cigarette smoking *	11 (3.2)	1 (0.4)	0.012
Alcohol drinking	22 (6.4)	14 (5.2)	0.529
Inadequate nutrition	82 (24.0)	82 (30.6)	0.134
Inadequate water intake	176 (51.5)	150 (56.0)	0.268

Data are expressed as numbers (percentages) or means ± SDs. Most of the categorical variables are binary, except for “BMD status.” * Fisher’s exact χ^2^ test for the data with insufficient expected cell counts. BMD, bone mineral density; NRT, number of remaining teeth.

**Table 3 ijerph-16-03945-t003:** Logistic regression analysis of factors associated with NRT <20.

Explanatory Variables	Model 1	Model 2	Model 3
	OR (95% CI)	OR (95% CI)	OR (95% CI)
Hypertension	1.28 (0.93–1.76)	§	§
Diabetes mellitus	1.45 (1.05–2.00) *	§	§
Hyperlipidemia	0.97 (0.65–1.44)	§	§
BMD status			
Normal ^¶^	–	–	–
Osteopenia	1.27 (0.87–1.84)	§	§
Osteoporosis	2.09 (1.32–3.31) *	§	§
Oral health related behaviors			
Tooth scaling infrequent	3.63 (2.27–5.80) *	3.11 (1.92–5.06) *	2.78 (1.70–4.56) *
Dental flossing infrequent	3.31 (2.11–5.20) *	2.38 (1.49–3.82) *	2.01 (1.24–3.26) *
Cigarette smoking	0.11 (0.01–0.88) *	§	§
Alcohol drinking	0.80 (0.40–1.60)	§	§
Inadequate nutrition	1.40 (0.98–2.00)	1.53(1.04–2.24) *	§
Inadequate water intake	1.20 (0.87–1.65)	§	§

Model 1: crude; Model 2: each variable was analyzed separately with adjustment by age and education; Model 3: a complete model with all variables including age and education. * *p* < 0.05. ^¶^ Reference group. § Variables were not selected into the model. Most of the categorical variables are binary, except for “BMD status.” NRT, number of remaining teeth; OR, odds ratio; CI, confidence interval; BMD, bone mineral density.

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
