# Peer review of "Factors Associated with Tooth Loss in Postmenopausal Women: A Community-Based Cross-Sectional Study"

_ijerph, 2019, doi:10.3390/ijerph16203945_

Round 1

Reviewer 1 Report

Can we blame menopause? Factors associated with tooth loss in postmenopausal women

Summary of manuscript: This manuscript reports the the role of oral hygiene in the number of remaining teeth in postmenopausal women. This report concludes that awareness of oral hygiene is the most important issue for retaining dentition in postmenopausal women. There are however several issues that needs to be addressed.

Results

The author performs a logistic regression analysis with 20 teeth as cut-off values. If authors want to examine the relationship between tooth loss and BMD etc, author should analyze the correlation analysis between these factors and add the results.

Discussion

Numerous past studies have reported the significant correlation between oral hygiene and tooth loss. Author should describe the novelty results revealed by this study.

Overall manuscript

Please proof-read the manuscript by a native English speaker.

Reviewer 2 Report

RE: Can we blame menopause? Factors associated with tooth loss in postmenopausal women

The manuscript aimed to explore the prevalence and factors related to NRT <20 among rural PMW, and to consider the role of BMD and oral health behaviors.

The manuscript is clearly written. My concern are as follows:

1) Title: The reviewer thinks this title does not represent the issue correctly. What does "Can we blame menopause?" mean?

2) Abstract: The aim is not written.

3) Line 44: "Retaining at least 20 natural teeth until 80 years of age has been defined as a key element of proper general health."

  It might be based on the geriatric research in Japan. However, the participants of this study were ≥50 years old, not 80 years old. Why did the authors set the cut-off as 20 teeth, even for the relatively younger generation.

4) The reason for tooth loss is mainly dental caries, periodontal disease, and tooth fracture.  So, it is possible that the women in postmenopausal stage lose their teeth due to dental caries or tooth fracture.  These causes are probably not related to postmenopausal situation. 

5) It is discouraging that the no information was collected with respect to actual oral hygiene status such as Oral Hygiene Index and/or Plaque Index that are commonly used in the epidemiological survey.

6) Line 89: “Does your diet include five food groups per day, namely grains, vegetables, milk, meat, and fruits?”; and “Do you drink at least 1,500 mL of water every day? 

   This questionnaire relates to the cause of tooth loss? If so, please put the references which scientifically prove the accuracy of this questionnaire.

7) The reviewer thinks that use of inter-dental brush is more effective for prevention of tooth loss in adults than dental floss use. However, there are no information about it.

Author Response

For reviewer 2:

Thank you for your comments.

Regarding the title does not represent the issue correctly.

Ans. We have reedited the title with red font on p1.

Regarding adding the aim in abstract.

Ans. We have reedited the aim in abstract on p1 with red font.

Regarding setting the cut-off of NRT as 20 teeth (Line 44), even for the age less than 65…

Ans. We had struggled for this cut-off number, such as 24-26 may be more reasonable for younger population. However, very few studies were found in the literatures. A campaign targeting the elderly to retain at least 20 teeth by the time they reach the age of 80 years (8020) has been in place since 1991 in Japan (Yamanaka, 2008). Therefore, NRT>20 has been recognized as an indicator of general health. Many important findings from the previous studies have also been targeted at a relatively younger generation of subjects with a threshold of 20 teeth (Hsu et al., 2009; Leroy & Declerck, 2013; Tsai et al., 2015). In order to uphold the principle that prevention is better than treatment, we set the cut-off as 20 teeth, even for the participants below 65 years old.

Hsu, K.-J., Yen, Y.-Y., Wu, Y.-M., Huang, R.-D., Lee, K.-T., Lei, Y.-N.,…Lee, H.-E. (2009). Relationship between the Number and Healthy Status of Remaining Teeth and Chewing Ability of the Middle-aged and Elderly. Taiwan Journal of Oral Medicine Sciences, 25(1), 64-77.  Leroy, R., & Declerck, D. (2013). Objective and subjective oral health care needs among adults with various disabilities. Clin Oral Investig, 17(8), 1869-1878. Tsai, S. J., Lin, M. S., Chiu, W. N., Jane, S. W., Tu, L. T., & Chen, M. Y. (2015). Factors associated with having less than 20 natural teeth in rural adults: a cross-sectional study. BMC Oral Health, 15, 158. Yamanaka, K., Nakagaki, H., Morita, I., Suzaki, H., Hashimoto, M., & Sakai, T. (2008). Comparison of the health condition between the 8020 achievers and the 8020 non-achievers. Int Dent J, 58(3), 146-150.

Regarding dental caries or tooth fractures might not related to the tooth loss in postmenopausal women, although dental caries, periodontal disease and dental fractures are the causes of tooth loss.

Ans. Yes, we also agree this comment.

Regarding without information of actual oral hygiene status such as oral hygiene index and/or plaque index.

Ans. This was a nurse-led health promotion program for exploring the tooth loss; the trained research assistants only assessed the number of teeth, not by the dentist. Therefore the oral hygiene related indicators were not included, which does limit the deeper findings of this study, and we have reedited more information in the section of limitations on p6 with red font. Thank you for your reminder, we will take this issue in mind and do further research in the future.

Regarding providing the references for proving the accuracy of the eating behaviors in questionnaire.

Ans. Previous studies have indicated that malnutrition is associated with dental and oral problems (Van Lancker et al., 2012; Wang, Chen, Liou, & Chou, 2014), a balance diet or multiple diet contributes to retaining tooth (Tanak et al., 2007; Yamanaka et al., 2008). A well-balanced diet is also advocated as a key factor in

promoting oral health (Daly & Smith, 2015; WHO, 2019). Therefore we adopted the reports from the WHO (2019) and the Taiwan Health Promotion Administration (2019). Healthy diet for adults should contain at least 400 g (five portions) of fruits and vegetables a day, nuts and whole grains, five groups of foods, and enough water instead of soft drinks as basis for questionnaire design. The citing references already indicated in p2 (line 53) and p3 (line 98). Other related literatures are as following.

WHO (2019). Oral health. Retrieved from https://www.who.int/news-room/fact-sheets/detail/oral-health  Taiwan Health Promotion Administration (2019). A well-balanced diet. Retrieved from https://www.hpa.gov.tw/Pages/List.aspx?nodeid=4086 Daly, B., & Smith, K. (2015). Promoting good dental health in older people: role of the community nurse. British journal of community nursing, 20(9), 431-436.   Van Lancker, A., Verhaeghe, S., Van Hecke, A., Vanderwee, K., Goossens, J., & Beeckman, D. (2012). The association between malnutrition and oral health status in elderly in long-term care facilities: A systematic review. International Journal of Nursing Studies, 49(12), 1568-1581.  Wang, T. F., Chen, Y. Y., Liou, Y. M., & Chou, C. (2014). Investigating tooth loss and associated factors among older Taiwanese adults. Arch Gerontol Geriatr, 58(3), 446-453.  Yamanaka, K., Nakagaki, H., Morita, I., Suzaki, H., Hashimoto, M., & Sakai, T. (2008). Comparison of the health condition between the 8020 achievers and the 8020 non-achievers. Int Dent J, 58(3), 146-150.  Tanaka, K., Miyake, Y., Sasaki, S., Ohya, Y., Matsunaga, I., Yoshida, T., . . . Oda, H. (2007). Relationship between intake of vegetables, fruit, and grains and the prevalence of tooth loss in Japanese women. Journal of Nutritional Science and Vitaminology, 53(6), 522-528.

Regarding information about more effective using interdental brush than flossing for prevention of tooth loss in adults.

Ans. Thank you for reminding. We have provided more information on p6 (line 196) with red font.

Round 2

Reviewer 1 Report

Factors associated with tooth loss in postmenopausal women: A community-based cross-sectional study

Summary of manuscript: This manuscript reports the the role of oral hygiene in the number of remaining teeth in postmenopausal women. This report concludes that awareness of oral hygiene is the most important issue for retaining dentition in postmenopausal women. There are however several issues that needs to be addressed.

This manuscript was well revised.

Reviewer 2 Report

The manuscript was revised correctly.